# Western blot using *Trypanosoma cruzi* chimeric recombinant proteins for the serodiagnosis of chronic Chagas disease: A proof-of-concept study

**Ramona Tavares Daltro[1], Emily Ferreira Santos[1], Ângelo Antônio Oliveira Silva[1], Natália Erdens Maron Freitas[1], Leonardo Maia Leony[1], Larissa Carvalho Medrado Vasconcelos[1], Alejandro Ostermayer Luquetti[2], Paola Alejandra Fiorani Celedon[3], Nilson Ivo Tonin Zanchin[4], Carlos Gustavo Regis-Silva[1], Fred Luciano Neves Santos[1,5]***

**1** Advanced Public Health Laboratory, Gonçalo Moniz Institute, Oswaldo Cruz Foundation, Salvador, Brazil, **2** Center of Studies for Chagas Disease, University Hospital, Federal University of Goiás, Goiânia, Brazil, **3** Laboratory of Molecular and Systems Biology of Trypanosomatids, Carlos Chagas Institute, Oswaldo Cruz Foundation, Curitiba, Brazil, **4** Structural Biology and Protein Engineering Laboratory, Carlos Chagas Institute, Oswaldo Cruz Foundation, Curitiba, Brazil, **5** Integrated Translational Program in Chagas disease from FIOCRUZ (Fio-Chagas), Oswaldo Cruz Foundation (FIOCRUZ-RJ), Rio de Janeiro, Rio de Janeiro, Brazil

* fred.santos@fiocruz.br

## Abstract

### Background

Chagas disease (CD) is caused by *Trypanosoma cruzi*. The chronic phase of CD is characterized by the presence of IgG anti-*T. cruzi* antibodies; and diagnosis is performed by serological methods. Because there is no reliable test that can be used as a reference test, WHO recommends the parallel use of two different tests for CD serodiagnosis. If results are inconclusive, samples should be subjected to a confirmatory test, e.g., Western blot (WB) or PCR. PCR offers low sensitivity in the chronic phase, whereas few confirmatory tests based on the WB method are commercially available worldwide. Therefore, new diagnostic tools should be evaluated to fill the gap in CD confirmatory tests. In recent years, four chimeric recombinant antigens (IBMP-8.1, IBMP-8.2, IBMP-8.3 and IBMP-8.4) have been evaluated in phase I, II and III studies using ELISA, liquid microarray and immunochromatography with 95–100% accuracy. Given the high diagnostic performance of these antigens, the present study investigated the ability of these molecules to diagnose chronic CD using a WB testing platform.

### Methodology/Principal findings

In this study, we analyzed the diagnostic potential of four chimeric antigens using 40 *T. cruzi*-positive, 24-negative, and three additional positive samples for visceral leishmaniasis (i.e., potentially cross-reactive) using WB as the diagnostic platform. Checkerboard titration with different dilutions of antigens, conjugated antigens, and serum samples was performed to standardize all assays. All IBMP antigens achieved 100% sensitivity, specificity, and accuracy, with the exception of IBMP-8.3, which had 100% specificity despite lack of

**Data Availability Statement:** All relevant data are within the manuscript and its Supporting Information files.

**Funding:** This work was supported by the Coordination of Superior Level Staff Improvement (CAPES; Finance Code 001 to FLNS, 88887.637869/2021-00 to LCMV, 88887.509223/2020-00 to LML, 88887.637641/2021-00 to RTD, and 88887.637758/2021-00 to NEMF), Research Support Foundation of the State of Bahia (FAPESB; BOL0543/2020 to ÂAOS and BOL0413/2021 to EFS), and Inova Fiocruz/VPPCB (FICORUZ; grant number VPPCB-008-FIO-18-2-20 to FLNS). NITZ and FLNS are research fellows at National Council for Scientific and Technological Development (CNPq; process no. 304167/2019-3 and 309263/2020-4, respectively). The funders had no role in study design, data collection and analysis, decision to publish, or preparation of the manuscript.

**Competing interests:** The authors have declared that no competing interests exist.

significance, but lower sensitivity (95%) and accuracy (96.9%). No cross-reactivity was observed in samples positive for leishmaniasis.

## Conclusions/Significance

The present phase I (proof-of-concept) study demonstrated the high diagnostic potential of these four IBMP antigens to discriminate between *T. cruzi*-positive and -negative samples, making them candidates for phase II and confirmatory testing with WB.

## Author summary

Chagas disease is a neglected infection that occurs mainly in Latin American countries. Infection occurs in several ways, the most common being vector infection by kissing bugs (family Triatominae). The disease progresses in two phases: acute and chronic. Since there is no reliable test that could serve as a gold standard, WHO recommends the parallel use of two tests. Conflicting results lead to an inconclusive diagnosis. In these cases, a confirmatory test is required, such as a Western blot (WB). However, few confirmatory tests based on the WB method are commercially available worldwide. Therefore, the aim of this study was to evaluate the use of four chimeric *T. cruzi* proteins (IBMP-8.1, IBMP-8.2, IBMP-8.3, and IBMP-8.4) developed by our group to distinguish *T. cruzi* positive and negative samples. Forty *T. cruzi*-positive samples, 24-negative, and three additional positive samples for visceral leishmaniasis were evaluated. All IBMP antigens achieved 100% sensitivity, specificity, and accuracy, with the exception of IBMP-8.3, which had 100% specificity, but lower sensitivity (95%) and accuracy (96.9%). No cross-reactivity was observed in positive samples for leishmaniasis. The present phase I (proof-of-concept) study demonstrated the high diagnostic potential of these four IBMP antigens to discriminate between *T. cruzi*-positive and -negative samples, making them candidates for phase II and confirmatory testing with WB.

## Introduction

*Trypanosoma cruzi*, the protozoan parasite causing Chagas disease (CD) or American trypanosomiasis, is not only a life-threatening neglected tropical disease in 21 Latin American countries, but also an emerging disease in non-endemic settings [1,2]. According to global estimates, 75 million people face risk of *T. cruzi* infection, with 5.7 million estimated cases of infection resulting in 7,500 deaths annually [1]. The classical route of infection occurs when infective metacyclic trypomastigotes, present in the feces and/or urine of triatomine vectors (also called "kissing bugs", cone-nosed bugs, and blood suckers), are inoculated through insect bites, skin abrasions, or the mucosa, such as the conjunctiva. Vector-borne transmission remains the predominant route in endemic areas, while other infection routes may be relevant in both endemic and non-endemic settings, e.g., oral transmission via the consumption of contaminated food or beverages, mother-to-child, blood-borne, bone marrow and organ-derived transmission, as well as infrequent laboratory accidents [3,4].

 CD progression is marked by two distinct phases. The initial acute period lasts for 60–90 days, during which individuals usually remain asymptomatic or present with a self-limiting febrile illness, characterized by fever, malaise, fatigue, body aches, headache, hepatosplenomegaly and atypical lymphocytosis. The main hallmark of this phase is an abundance of

circulating parasites, which can easily be detected by direct microscopic examination. In 5–10% of symptomatic cases, fatality occurs due to acute CD as a result of complications associated with meningoencephalitis and/or myocarditis [3]. If left untreated, individuals then enter a long-lasting chronic phase. Parasite replication becomes controlled by cell-mediated immune response, symptoms resolve spontaneously, and parasites are undetectable by direct parasitological methods. While most individuals remain asymptomatic, infection persists throughout their lifetime. It has been estimated that 20–30% of chronically infected individuals may develop, over a course lasting from years to decades, debilitating conditions, such as severe cardiac, digestive (typically enlargement of the esophagus or colon) or neurological complications, or mixed alterations [3,5,6].

Laboratory diagnosis during the chronic phase is based on the detection of anti-*T. cruzi* antibodies by indirect immunoassays, including enzyme-linked immunosorbent assay (ELISA), indirect immunofluorescence (IIF), indirect hemagglutination (IHA), and chemiluminescent (CLIA) methods. Despite the commercial availability of several methodologies, operational and technical issues hamper the performance of serological assays, which has been attributed to variations in disease prevalence [7,8], the choice of antigens employed to sensitize the solid phase of immunoassays [9], variable immune responses in *T. cruzi*-infected individuals [10], and extensive *T. cruzi* genetic and phenotypic intraspecific diversity [11]. Accordingly, the WHO advises the concomitant use of two different serological assays to achieve a definitive diagnosis [12]. Chimeric antigens, composed of linear immunodominant conserved and repetitive epitopes of structural, surface and cytosolic *T. cruzi* antigens, offer increased epitope diversity, and have thusly been proposed as an appropriate tool to enhance immunoassay diagnostic parameters [13–15]. In recent years, the diagnostic performance of four *T. cruzi* chimeric proteins (IBMP-8.1, IBMP-8.2, IBMP-8.3, and IBMP-8.4) has been comprehensively assessed in both endemic and non-endemic settings across South America [16–20], as well as in Barcelona (Spain) [21], with high performance noted despite variability in sample geographic origin. Moreover, all four antigens performed remarkably well when utilized for the serodiagnosis of chronic CD in settings where *Leishmania* and *T. cruzi* are considered co-endemic [22].

Following WHO recommendations, around 5% of samples submitted to dual-serological assay testing return discordant or doubtful results (values falling within the cutoff range or indeterminate zone). Some studies have reported similar values ranging from 2.9% to 3.3% [23,24]. While initial testing can be repeated, in some cases a third, preferably confirmatory test has also been used [25], e.g., radioimmunoprecipitation assays (RIPA), immunoblot assays using recombinant antigens, and Western blot assays employing trypomastigote excreted-secreted antigens (TESA-blot; bioMérieux, Brazil), native trypomastigote and amastigote antigens from the CL Brener *T. cruzi* strain (Chagas Western Blot IgG assay—Chagas blot; LDBio Diagnostics, Lyon, France) or a set of *T. cruzi* recombinant antigens: CRA, FRA, TcD, MAP, SAPA, Ag39 and Tc24 (HBK 740 Immunoblot Linhas anti-*T. cruzi*, EMBRABIO, São Paulo-SP, Brazil) and FP10, FP6, FP3 and TcF (Abbott ESA Chagas, Abbott Laboratories, IL, USA). RIPA, a highly complex technique, is labor-intensive, expensive and involves radioactivity, which limits its use [26,27]. INNO-LIA Chagas (Innogenetics N.V., Ghent, Belgium) is a commercially available immunoblot assay consisting of seven recombinant and synthetic antigens: CRA, FRA, Tc-24, SAPA, MAP, TcD and Ag39, which are coated at separate locations onto a single nylon membrane strip. While offering high diagnostic performance, results can be difficult to interpret, and this platform remains expensive [28–31]. Until 2015 and 2017, respectively, HBK 740 Immunoblot Linhas anti-*T. cruzi* and TESA-blot were routinely used for confirmatory testing [32]; however, commercial productions have since been discontinued, leaving a gap in the performance of confirmatory chronic CD diagnosis. However, two other

WB are commercially available worldwide. The Chagas blot and Abbott ESA Chagas, whose use is limited to Europe and the USA, respectively. The high diagnostic performance offered by IBMP chimeric antigens and the need for confirmatory serological methods led us to perform a proof-of-concept study (phase I) to evaluate the capability of these antigens to detect anti-*T. cruzi* IgG using a Western blot platform in serum samples obtained from distinct clinical presentations, representative of diverse Brazilian geographical settings.

## Material and methods

### Ethics statement

This study was approved by the Institutional Review Board for Human Research at the Gonçalo Moniz Institute (IRB/IGM/Fiocruz-BA), Salvador-Bahia, Brazil (protocol no. 67809417.0.0000.0040). To protect patients' privacy, the IRB required that all samples be coded to mask identification, thereby avoiding the need to obtain verbal or written consent. Data of all patients were fully anonymized prior to researcher access.

### Recombinant protein synthesis

The synthesis of the four IBMP chimeric proteins was performed as described previously [19]. Briefly, synthetic genes were obtained from a commercial supplier (GenScript, Piscataway, NJ, USA) and subcloned into the pET28a vector. The antigens were expressed in *Escherichia coli*-Star (DE3) cells that had grown in Luria-Bertani medium supplemented with 0.5 mM IPTG (isopropyl-β-D-1-thiogalactopyranoside). His-tagged chimeric antigens were purified by ion exchange and affinity chromatography, then quantified by fluorometry (Qubit 2.0, Invitrogen Technologies, Carlsbad, CA, USA) in accordance with the manufacturer's protocol.

### Western blot assay

Purified IBMP proteins were individually dissolved in Laemmli buffer and separated on a 15% denaturing polyacrylamide gel. Tris-glycine based-buffer (25 mM Tris/250 mM glycine/0.1% SDS) was used as a running buffer. Proteins were transferred to a 0.45 μm pore-size nitrocellulose membrane (Amersham Protran NC Nitrocellulose Membrane, GE Healthcare, USA) using a semidry apparatus (Trans-Blot SD System, Bio-Rad Laboratories, USA) at 20 volts for 1 hour. Towbin buffer was used as a transfer buffer. Following blotting, the membrane was blocked in Phosphate-Buffered Saline (PBS) containing 0.05% (v/v) Tween 20 and 5% skim dry milk (blocking solution). The membrane was placed on a miniblotter (Miniblotter 25, Immunetics, USA) with 25 channels (4.0 mm each). Serum samples were diluted in blocking solution and then incubated with the membrane for 1 hour. The membrane was washed three times with PBS containing 0.05% (v/v) Tween 20 (washing solution). Horseradish peroxidase (HRP)-conjugated goat anti-human IgG (Bio-Manguinhos, Fiocruz, Rio de Janeiro, RJ, Brazil) was diluted in blocking solution and then incubated with the membrane for 1 hour. After washing three times with washing solution, the membrane was washed once with PBS. Pierce™ ECL Western Blotting Substrate (Thermo Scientific™, Waltham, MA, USA) was then added to the membrane. Bands were read on an ImageQuant LAS 4000 series biomolecular imaging system (General Electric, Boston, MA, USA) and images were saved in tagged image file (.tiff) format.

### Sampling

The necessary sample size was calculated using OpenEpi open-source software [33] considering an infinite population, expected sensitivity and specificity of 99%, a 95% confidence interval, and an absolute error of 5%. Based on these specifications, the minimum sample required

to implement the present study was estimated at 32 samples: 16 sera each from *T. cruzi*-infected and uninfected individuals. Anonymized human sera were obtained from *T. cruzi*-infected (n = 38) and uninfected (n = 24) individuals residing in CD endemic and nonendemic areas in the following Brazilian states: Alagoas, Amapá, Bahia, Goiás, Minas Gerais, Paraíba and Pernambuco (Fig 1). All samples were recharacterized based on negativity or positivity using two of the following serological CD tests: ELISA Chagas III (BIOSChile, Ingeniería Genética S.A., Santiago, Chile), Imuno-ELISA Chagas (Wama Diagnóstica, São Paulo, Brazil), Gold ELISA Chagas (REM Diagnóstica, São Paulo, Brazil), or Liaison XL Murex Chagas (DiaSorin, Saluggia, Italy). Samples with discordant results or those judged to be inconclusive were excluded. Each sample was assigned a numeric code in the laboratory to ensure a blinded analysis.

In addition, two WHO International Standards (IS) (IS 09/186 and IS 09/188) were also included; these sera are representative of seropositive samples from autochthonous individuals living in Brazil, a region where *T. cruzi* discrete typing unit (DTU) TcII is predominant, and Mexico, where *T. cruzi* DTU TcI predominates, respectively [34]. *T. cruzi*-positive samples were obtained from individuals with different clinical presentations of chronic CD, including indeterminate, cardiac, and mixed forms (Fig 1). To evaluate cross-reactivity, three serum samples from individuals serologically diagnosed with visceral leishmaniasis were also included. Serum samples were stored in sealed well-labeled microtubes at -20˚C until immunoassays were performed.

## Image analysis

Assay results were interpreted by captured images of membrane strips on an ImageQuant LAS 4000 digital imaging system (General Electric, Boston, MA, USA) after 30 seconds of exposure to the light produced during the chemiluminescence reaction. Monochrome image data was saved in.tiff format, and inverted-color images were obtained using ImageJ (National Institutes of Health—NIH, Bethesda, MD, USA; https://imagej.nih.gov/ij/) [35]. Regions of interest, i.e., either background area (no reaction) or test lines, were selected using the rectangular tool on the ImageJ menu bar. Identically sized areas (containing the same number of pixels) were selected in each acquired image. Next, the areas under the peak values obtained from test lines and background areas were calculated by virtual chromatogram in ImageJ software. After subtracting background pixel intensity from test strip measurements, the final values were transferred to a software program to perform statistical analysis.

## Statistical analysis

Statistical testing was performed using GraphPad version 8 (GraphPad Prism, San Diego, CA, USA) computer graphing software. Variables of interest were expressed as geometric means ± SD. The Shapiro-Wilk test was employed to test data normality. Wilcoxon-Mann-Whitney or Kruskal-Wallis tests were used when the null hypothesis was rejected, while the Student's T-test was employed when data normality was confirmed. All analyses were two-tailed, and a p-value less than 5% ($p < 0.05$) was considered significant. Cut-off point analysis was used to identify the optimal pixel intensity that differentiated between *T. cruzi*-positive and negative samples. Threshold values were established based on the area under ROC curve (AUC). Assay results were normalized by determining the relative band intensity (RBI) representative of the ratio between each sample's final pixel intensity and its respective cutoff value. Samples that returned RBI < 1.00 were considered negative. Samples were deemed inconclusive when RBI values fell into an indeterminate zone: $0.90 \leq RBI \leq 1.10$. AUC results were used to evaluate the global accuracy for each IBMP antigen, classified as outstanding (1.00), elevated (0.99–0.82), moderate (0.81–0.62), or low (0.61–0.51) [36]. WB performance

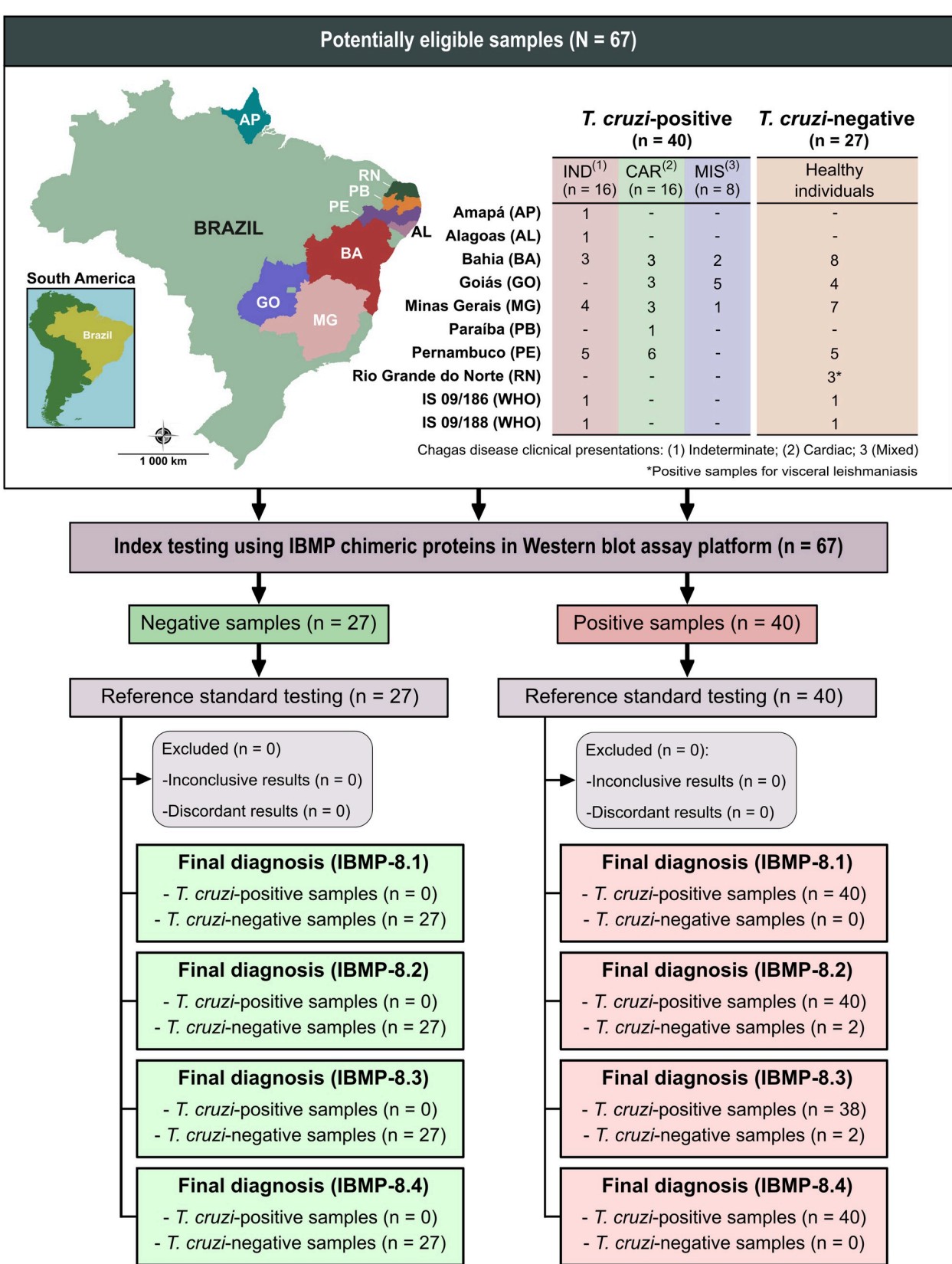

**Fig 1. Flowchart illustrating study design in conformity with the Standards for Reporting of Diagnostic Accuracy Studies (STARD) guidelines.** Source base layer and credit base layer: https://data.humdata.org/ published under creative commons attribution for intergovernmental organizations: https://data.humdata.org/dataset/geoboundaries-admin-boundaries-for-brazil.

parameters were determined using a dichotomous approach regarding sensitivity (Sen), specificity (Spe) and accuracy (Acc). Confidence intervals at 95% (95% CI) were calculated to reflect the precision of the estimates obtained, and the absence of overlapping 95% CI values was considered indicative of statistical significance. Cohen's *kappa* (κ) analysis was used to determine the strength of agreement between the standard reference tests and WB, interpreted as follows: perfect agreement (k = 1.0), almost perfect agreement ($0.81 \leq k < 1.0$), substantial agreement ($0.61 \leq k < 0.80$), moderate agreement ($0.41 \leq k < 0.60$), fair agreement ($0.21 \leq k < 0.40$), slight agreement ($0 < k < 0.20$), and poor agreement (k = 0) [37]. A flowchart (Fig 1) and a checklist (S1 Checklist) have been provided in accordance with the Standards for Reporting of Diagnostic Accuracy Studies (STARD) guidelines [38].

## Results

### Western blot standardization

The optimal dilutions of IBMP antigens, sera and antibody-enzyme conjugate were evaluated by checkerboard titration. Optimal conditions were selected by considering the highest differences between median RBI values for *T. cruzi*-positive and *T. cruzi*-negative samples. Accordingly, experimental conditions were classified as satisfactory when sera were diluted at 1:100, the antibody-enzyme conjugate was diluted at 1:2,000, and 12.5 ng of each IBMP was used to sensitize nitrocellulose strips.

### Phase I study

The phase I study (proof-of-concept) was carried out employing a serological panel composed of samples from *T. cruzi*-positive and *T. cruzi*-negative individuals from both endemic and non-endemic settings. Importantly, several clinical CD presentations were considered by the present study (Fig 1). Based on AUC values, all IBMP chimeric antigens were classified as having outstanding diagnostic potential (Fig 2). IBMP-8.1, IBMP-8.2 and IBMP-8.4 produced a sensitivity of 100%, while IBMP-8.3 offered 95%. All IBMP chimeric antigens attained the maximum specificity value of 100%, and maximum accuracy was achieved when samples were assayed using either IBMP-8.1, IBMP-8.2 or IBMP-8.4. Assays employing IBMP-8.3 exhibited an accuracy of 96.9% (Fig 2). No differences were seen among the four IBMP chimeric antigens in terms of AUC, sensitivity, specificity, or accuracy at 95% CI. Cohen's *Kappa* values indicated perfect (IBMP-8.1, IBMP-8.2 and IBMP-8.4) and almost perfect (IBMP- 8.3) agreement with the reference tests.

The highest RBI values were obtained for *T. cruzi*-positive samples using IBMP-8.2 (7.36) and IBMP-8.4 (7.32), followed by IBMP-8.3 (4.13) and IBMP-8.1 (3.76). Considering the lack of overlapping 95% CI values, significative differences were observed between IBMP-8.1 and IBMP-8.4. With regard to *T. cruzi*-negative samples, the lowest RBI values were seen for IBMP-8.3 (0.05), IBMP-8.4 (0.13) and IBMP-8.2 (0.25), while IBMP-8.1 produced the highest value (0.37), yet without any significative differences. Significantly superior mean RBI values were obtained from *T. cruzi*-positive samples compared to negative samples (p < 0.0001) using all four investigated antigens. No *T. cruzi*-positive or negative samples yielded inconclusive results that fell in the indeterminate zone (RBI = 1.0 ± 10%) (Fig 2; individual RBI values are available in S1 Table). Visual analysis of the test strips, as shown in Fig 3 for *T. cruzi*-

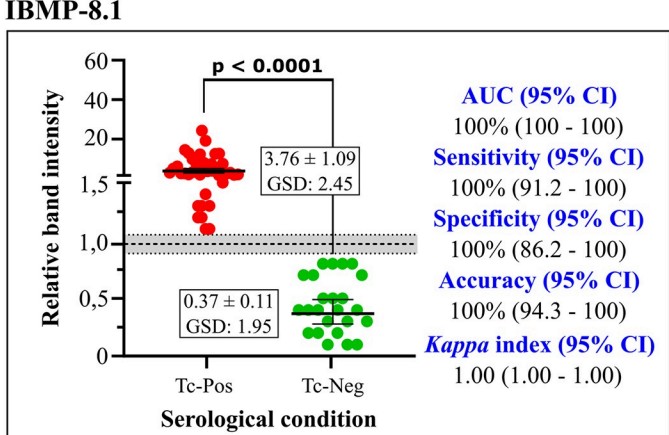

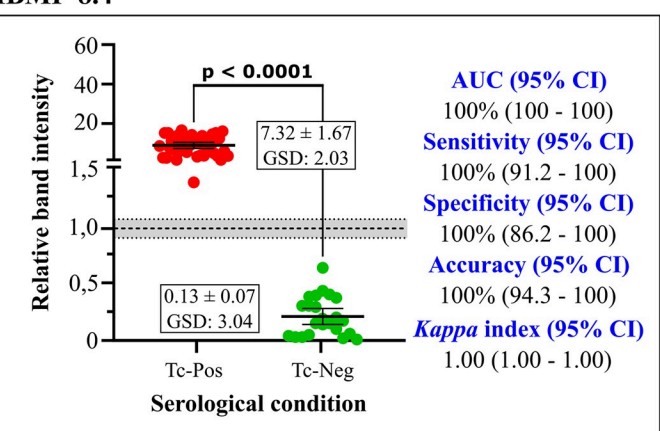

**Fig 2. Relative band intensity (RBI) and diagnostic performance parameters obtained from *Trypanosoma cruzi*-positive (Tc-Pos) and *T. cruzi*-negative (Tc-Neg) serum samples.** The cut-off value was set at a reactivity index value of 1.0, with the shaded area representing the grey zone (i.e., an indeterminate result; RI = 1.0 ± 0.10). Horizontal lines and numbers for each result group represent geometric means (± 95%CI). GSD represents geometric standard deviation.

positive samples and in Fig 4 for *T. cruzi*-negative samples, revealed the presence of visible bands in all positive samples, with no bands seen in any of the *T. cruzi*-negative samples regardless of the protein employed. In general, IBMP-8.3 generated less easily evidenced signals compared to IBMP-8.4.

## Cross-reactivity assessment

A panel composed of three serum samples was used to evaluate cross-reactivity with antibodies produced against visceral leishmaniasis. As illustrated in Fig 5, no cross-reactions were observed when assaying leishmaniasis-positive serum samples using all four chimeric proteins. A faint band was observed when sample #03 was assayed with IBMP-8.4; however, this sample returned an RBI signal below the established cut-off. Mean RBI values ranged from 0.02 for IBMP-8.1, 0.21 for IBMP-8.2 and 0.02 for IBMP-8.3 to 0.02 for IBMP-8.4.

## Discussion

The Western blot method is commonly used for the clinical diagnosis of several parasitic and fungal diseases, in addition to providing confirmatory diagnosis of infections caused by HIV-

## *Trypanosoma cruzi*-positive samples

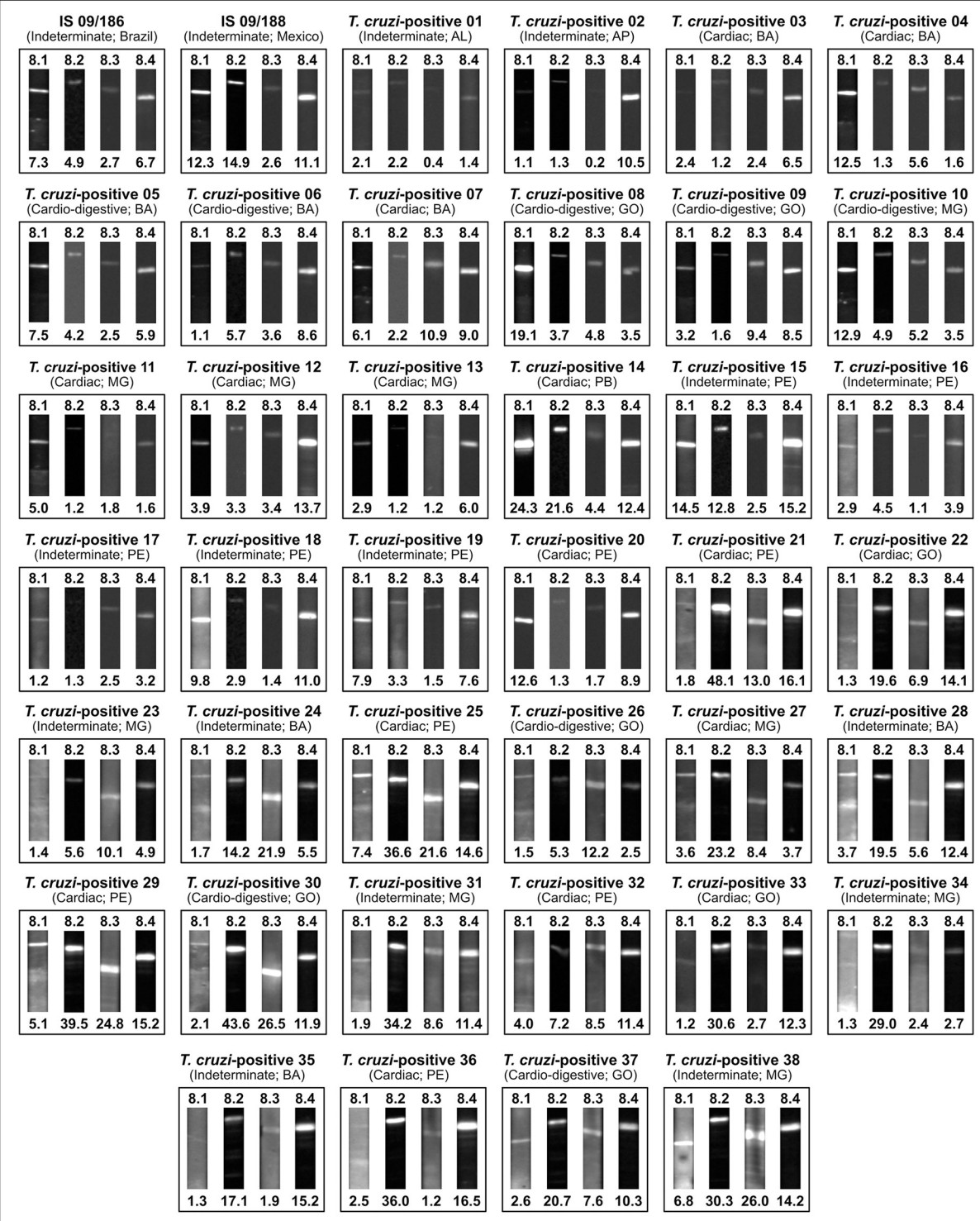

**Fig 3. Photographic images of nitrocellulose test strips result from *Trypanosoma cruzi*-positive samples.** Relative band intensity (RBI) values respective to each IBMP antigen are shown below the images pertaining to each sample. Abbreviations: AL (Alagoas); AP (Amapá); BA (Bahia); GO (Goiás); MG (Minas Gerais); PE (Pernambuco).

## *Trypanosoma cruzi*-negative samples

**Fig 4. Photographic images of nitrocellulose test strips result from *Trypanosoma cruzi*-negative samples.** Relative band intensity (RBI) values respective to each IBMP antigen are shown below the images pertaining to each sample. Abbreviations: AL (Alagoas); AP (Amapá); BA (Bahia); GO (Goiás); MG (Minas Gerais); PE (Pernambuco).

1/2 and HTLV-1/2 [39–41]. Several researchers have investigated the usefulness of WB in the confirmatory diagnosis of chronic CD when conventional tests returned inconclusive results, reporting high values of sensitivity and specificity [42–45]. The development of a methodology to clarify results that are inconclusive after using conventional assays is extremely necessary because at least 50% of discordant results are from individuals affected by Chagas disease [23]. In 1986, a study used WB to assess the performance of epimastigote antigens in diagnosing chronic CD; however, the low purity of the antigens employed resulted in cross reactions against anti-*Leishmania braziliensis* and anti-*Leishmania donovani* [46]. On the other hand, Reiche and colleagues alternatively evaluated antigenic polypeptides using the WB platform, obtaining conclusive results capable of confirming laboratory-based chronic CD diagnosis [43]. Accordingly, in combination with purified or recombinant antigens, WB is still considered a promising strategy for diagnostic confirmation. The present study assessed the diagnostic potential of four *T. cruzi*-chimeric antigens using the WB diagnostic platform; all antigens presented maximum values (98.5–100%) under AUC analysis, indicating the outstanding

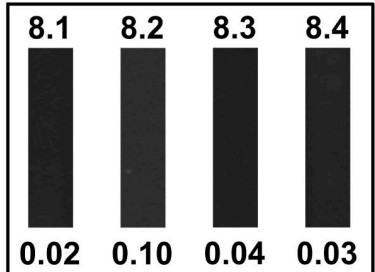



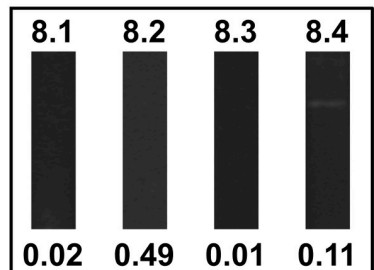

|  | IBMP-8.1 | IBMP-8.2 | IBMP-8.3 | IBMP-8.4 |
|---|---|---|---|---|
| **RBI** | 0.02 | 0.15 | 0.03 | 0.05 |
| **False-positive (%)** | 0 | 0 | 0 | 0 |
| **Specificity (%)** | 100 | 100 | 100 | 100 |
| ***Kappa* index (IC95%)** | 1.00 | 1.00 | 1.00 | 1.00 |

**Fig 5. Analysis of cross-reactivity using chimeric *Trypanosoma cruzi* proteins to diagnose sera from visceral leishmaniasis patients.** Relative band intensity (RBI) values for each sample are shown below the images.

ability of these antigens to differentiate between *T. cruzi*-positive and *T. cruzi*-negative samples, in the absence of cross-reactivity among visceral leishmaniasis-positive sera.

TESA (Trypomastigote Excreted-Secreted Antigens) is a complex mixture of antigenic molecules, composed mainly of transialidases. This antigen mixture strongly stimulates the humoral response of infected individuals, and has been largely used in investigations involving the diagnosis of chronic CD via WB, resulting in high sensitivity and specificity [32,47]. Based on the literature, BioMérieux SA do Brasil started to commercialize the WB-based TESAcruzi test using TESA as an antigenic preparation in 2004. This diagnostic platform was authorized for use in Brazil in this same year, and eventually worldwide as a confirmatory serological test for chronic CD, achieving recommended status for this purpose by the 2nd Brazilian Consensus on Chagas Disease [48]. However, the discontinuation of production resulted in a gap with respect to confirmatory CD diagnostic testing.

Similar to the TESAcruzi, the Chagas Western blot IgG assay (Chagas blot) uses a complex mixture of antigenic molecules to coat a nylon strip. This assay is manufactured by a French company (LDBio Diagnostics) and uses native trypomastigote and amastigote antigens from the CL Brener *T. cruzi* strain (TcIV). In 2021, a study using samples from *T. cruzi*-positive individuals living in endemic areas of Argentina [49] showed that this kit has great potential for use as a confirmatory test (sensitivity and specificity of 100%). However, the authors note that these results must be confirmed in larger tests with many sera from different regions of South America before this immunoblot can be considered a universal confirmatory test.

An important pitfall of TESAcruzi, which limited its use, was a high rate of false-negativity. One strategy to overcome this obstacle is based on the use of recombinant or chimeric recombinant antigens. The use of chimeric recombinant proteins, in which a single molecule displays several epitopes from different polypeptides simultaneously, has improved the diagnostic

performance of immunoassays [13,50]. Furthermore, greater commercial viability is obtained, as a greater number of sequences are generated from a single production process. The availability of multiple epitopes also increases assay sensitivity, as the same amount of surface area will therefore capture a greater number of antibodies from each sample [51–53]. However, to date few studies have evaluated the use of chimeric proteins as an antigenic matrix in a WB-based diagnostic immunoassay platform. In 2010, the Abbott ESA Chagas, an immunoblot based on 4 chimeric recombinant *T. cruzi* antigens (FP10, FP6, FP3, and TcF) [52], was evaluated using samples from different groups of *T. cruzi* infected and uninfected individuals. The high clinical and analytical sensitivity values as well as the simplicity of the method led the authors to conclude that the Abbott ESA Chagas could replace RIPA as the confirmatory test of choice for the detection of antibodies to *T. cruzi* [54]. In November 2011, the Abbott ESA Chagas was approved by the US FDA for the confirmation of blood donors who are repeatedly reactive in Chagas screening tests. The diagnostic performance obtained here using WB corroborates our previous findings obtained using a variety of diagnostic platforms [16–22,55–58]. Indeed, all four chimeric proteins achieved 100% sensitivity and specificity, with the exception of IBMP-8.3 (95% sensitivity).

TESAcruzi has been considered a suitable confirmatory technique due to high sensitivity and specificity [59], similar to the diagnostic performance demonstrated by the IBMP antigens studied herein. However, some factors associated with the production of TESAcruzi are relevant to consider, such as high cost, laborious technique and the amount of time and resources required to cultivate pathogenic parasites, making the use of IBMP chimeric recombinant antigens a promising alternative strategy for confirmatory chronic CD diagnosis. Obtained using genetic engineering tools, IBMP chimeras are designed to provide the physicochemical properties necessary to improve industrial processes and thus lower manufacturing costs while maintaining elevated diagnostic potential. Furthermore, the use of chimeric proteins allows laboratories to reduce the amount of antigens needed in diagnostic assays [50]. The results obtained herein clearly demonstrate that all four molecules efficiently and safely differentiate between seronegative and seropositive samples, making all four eligible for phase II evaluation, which we plan to carry out as a continuation of the present study.

In the scientific literature, WB assay conditions vary widely in terms of the amount of antigen used, as well as conjugate and serum sample dilutions. The amount of antigen used for sensitization ranges from 0.2 μg to 500 μg [60–63], which can be prepared from parasite extracts or the use of recombinant antigens. For the present WB assay employing IBMP proteins, a smaller amount of antigen was used: 12.5 ng/4 mm for each of the four proteins. As the TESAcruzi package insert provided no information regarding the amount of TESA antigen used in each test strip, studies report highly variable dilutions of conjugated antibodies, ranging from 1:200 to 1:4,000 [32,64,65]. TESAcruzi employed anti-human IgG conjugated to peroxidase (1 ml per test), in the absence of a specified antibody dilution. In the WB-IBMP evaluated herein, the conjugate was standardized at a 2,000-fold dilution, requiring only 400 μL per reaction. With regard to the serum dilution, both WB-IBMP and TESAcruzi require similar dilutions (1:100 for both assays).

One of the main advantages of WB assays is a low incidence of cross-reactions, since this technique allows for the identification of antibodies against different polypeptide fractions of parasite antigens, which present specific band patterns [44,66]. Although cross-reactivity with species of the *Leishmania* genus present significant limitations for conventional serological diagnosis of chronic CD [22,46,67], this is largely mitigated when using WB. In 2019, we evaluated cross-reactivity by ELISA between the four IBMP antigens using serum samples from individuals positive for cutaneous (n = 600) and visceral (n = 229) leishmaniasis [22]. For cutaneous leishmaniasis, IBMP-8.1 and IBMP-8.2 presented reactivity in just 0.70% of samples,

while IBMP-8.3 was reactive against 0.35% of the tested samples. For visceral leishmaniasis, cross-reactivity was observed in 3.49% and 0.58% of samples evaluated by IBMP-8.2 and IBMP-8.3, respectively. No cross-reactions were detected in any of the 829 samples tested by IBMP-8.4. Similar results were observed for IBMP-8.1 molecule when evaluating visceral leishmaniasis-positive samples. Herein, no cross-reactivity was seen in three positive samples for visceral leishmaniasis evaluated using WB-IBMP. The phase II study will expand the number of samples used to evaluate cross-reactivity more comprehensively.

The present study (phase I) represents the first attempt to assess the use of chimeric recombinant proteins using the WB platform. The results obtained herein demonstrate the high discriminatory capacity of all four IBMP antigens to efficiently and safely differentiate *T. cruzi*-positive from -negative samples; however, a phase II study involving an increased number of samples will serve to further elucidate and confirm the presently described findings.

## Supporting information

**S1 Checklist. STARD checklist.** Standards for the Reporting of Diagnostic Accuracy Studies (STARD) checklist for reporting of studies of diagnostic accuracy.
(DOCX)

**S1 Table. Individual relative band intensity points for diagnostic performance assessment.**
(XLSX)

## Acknowledgments

The authors would like to thank Andris K. Walter for critical analysis, English language revision and manuscript copyediting assistance. We also express our gratitude to Pedro Albajar Viñas for providing the two WHO International Standards (IS) used in our study.

## Author Contributions

**Conceptualization:** Alejandro Ostermayer Luquetti, Nilson Ivo Tonin Zanchin, Fred Luciano Neves Santos.

**Data curation:** Carlos Gustavo Regis-Silva, Fred Luciano Neves Santos.

**Formal analysis:** Ramona Tavares Daltro, Natália Erdens Maron Freitas, Paola Alejandra Fiorani Celedon, Nilson Ivo Tonin Zanchin, Carlos Gustavo Regis-Silva, Fred Luciano Neves Santos.

**Funding acquisition:** Nilson Ivo Tonin Zanchin, Fred Luciano Neves Santos.

**Investigation:** Ramona Tavares Daltro, Emily Ferreira Santos, Ângelo Antônio Oliveira Silva, Natália Erdens Maron Freitas, Leonardo Maia Leony, Larissa Carvalho Medrado Vasconcelos, Paola Alejandra Fiorani Celedon, Carlos Gustavo Regis-Silva, Fred Luciano Neves Santos.

**Methodology:** Ramona Tavares Daltro, Emily Ferreira Santos, Ângelo Antônio Oliveira Silva, Natália Erdens Maron Freitas, Leonardo Maia Leony, Larissa Carvalho Medrado Vasconcelos, Paola Alejandra Fiorani Celedon, Carlos Gustavo Regis-Silva, Fred Luciano Neves Santos.

**Project administration:** Carlos Gustavo Regis-Silva, Fred Luciano Neves Santos.

**Resources:** Fred Luciano Neves Santos.

**Software:** Fred Luciano Neves Santos.

**Supervision:** Carlos Gustavo Regis-Silva, Fred Luciano Neves Santos.

**Validation:** Ramona Tavares Daltro, Fred Luciano Neves Santos.

**Visualization:** Fred Luciano Neves Santos.

**Writing – original draft:** Ramona Tavares Daltro, Emily Ferreira Santos, Ângelo Antônio Oliveira Silva, Natália Erdens Maron Freitas, Leonardo Maia Leony, Larissa Carvalho Medrado Vasconcelos, Paola Alejandra Fiorani Celedon, Carlos Gustavo Regis-Silva, Fred Luciano Neves Santos.

**Writing – review & editing:** Ramona Tavares Daltro, Leonardo Maia Leony, Alejandro Ostermayer Luquetti, Paola Alejandra Fiorani Celedon, Nilson Ivo Tonin Zanchin, Carlos Gustavo Regis-Silva, Fred Luciano Neves Santos.

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
