## [Decision Letter · Decision Letter 0]

30 Sep 2022

Dear PhD Santos,

Thank you very much for submitting your manuscript "Western blot using Trypanosoma cruzi chimeric recombinant proteins for the serodiagnosis of chronic Chagas disease: a proof-of-concept study" for consideration at PLOS Neglected Tropical Diseases. As with all papers reviewed by the journal, your manuscript was reviewed by members of the editorial board and by several independent reviewers. The reviewers appreciated the attention to an important topic. Based on the reviews, we are likely to accept this manuscript for publication, providing that you modify the manuscript according to the review recommendations. 

In addition to observations of the three reviewers, we ask you to pay attention to the following points during revision process:

- probably your are not aware about the commercially available (at least in Europe) LDBio western blot for Chagas disease; reviewer 2 is giving you the reference; please revise the paper taking it into account and add the reference;

- in the introduction and discussion give more notes on the problem of discordance in Chagas disease diagnosis (ref http://link.springer.com/10.1007/s10096-011-1393-9) taking into account the concept of the at least 50% rate of Chagas disease affected individuals among the discordants (see Moure et al DOI:https://doi.org/10.1016/j.cmi.2016.06.001, DOI: 10.1016/j.actatropica.2018.05.010)

- in the conclusions, clearly state that this is a phase I study and a larger sample study is needed to confirm results.

Sincerely,

Andrea Angheben

Academic Editor

Charles Jaffe

Section Editor

Dear authors, in addition to observations of the three reviewers, I ask you to pay attention to the following points during revision process:

- probably your are not aware about the commercially available (at least in Europe) LDBio western blot for Chagas disease; reviewer 2 is giving you the reference; please revise the paper taking it into account and add the reference;

- in the introduction and discussion give more notes on the problem of discordance in Chagas disease diagnosis (ref http://link.springer.com/10.1007/s10096-011-1393-9) taking into account the concept of the at least 50% rate of Chagas disease affected individuals among the discordants (see Moure et al DOI:https://doi.org/10.1016/j.cmi.2016.06.001, DOI: 10.1016/j.actatropica.2018.05.010)

- in the conclusions, clearly state that this is a phase I study and a larger sample study is needed to confirm results.

Reviewer's Responses to Questions

**Key Review Criteria Required for Acceptance?**

**Methods**

-Are the objectives of the study clearly articulated with a clear testable hypothesis stated?

-Is the study design appropriate to address the stated objectives?

-Is the population clearly described and appropriate for the hypothesis being tested?

-Is the sample size sufficient to ensure adequate power to address the hypothesis being tested?

-Were correct statistical analysis used to support conclusions?

-Are there concerns about ethical or regulatory requirements being met?

Reviewer #1: The study was designed and carried out nicely. There is no concern at all on the whole aspect of the study

Reviewer #2: This is an interesting paper about the possible use of four chimeric proteins to have a western-blot reference test for the diagnosis of T. cruzi infection.

The study design is appropriate, the samples clearly described.

The sample size is small but this paper is only a phase I (proof of concept), it is acceptable.

Reviewer #3: The objective of the study were clearly articulated with a clear testable hypothesys stated

The study design is appropriate

The population is clearly described and appropriated

The sample size is enough for this study

I agree with the chisen statistical methods

No concerns about ehic problems or so on

**Results**

-Does the analysis presented match the analysis plan?

-Are the results clearly and completely presented?

-Are the figures (Tables, Images) of sufficient quality for clarity?

Reviewer #1: The results were recorded using a non-human reader to prevent bias. The analysis was carried out well using the best statistical calculation.

Reviewer #2: The results are clearly exposed, with figures of sufficient quality

Reviewer #3: The analysis match the analysis plan

Results are shown clearly

OK for figures

**Conclusions**

-Are the conclusions supported by the data presented?

-Are the limitations of analysis clearly described?

-Do the authors discuss how these data can be helpful to advance our understanding of the topic under study?

-Is public health relevance addressed?

Reviewer #1: Conclusions were based on the study results. The study will fill the gap in the lack of confirmatory chronic Chagas disease

Reviewer #2: The conclusions are supported by the data.

These results need to be conforted by further studies, and the cross reactions studied on a larger sample size.

Reviewer #3: Conclusions are supported by the data presented

Limit of the work are described

It is relevant for the Public Health

**Editorial and Data Presentation Modifications?**

Reviewer #1: None

Reviewer #2: The authors say (line 56) : there is no reliable test that could serve as a gold standard.

there are published data (pathogens 2021 Nov 10;10(11):1455) about a commercialised CE marketed western blot for the diagnosis of Chagas disease 

maybe it would be interesting to add this paper in the references and to take it in charge in the discussion

Reviewer #3: No suggestion

**Summary and General Comments**

Reviewer #1: None

Reviewer #2: (No Response)

Reviewer #3: No comment: good work.

PLOS authors have the option to publish the peer review history of their article (what does this mean?). If published, this will include your full peer review and any attached files.

Reviewer #1: Yes: Sukwan Handali

Reviewer #2: No

Reviewer #3: Yes: Romualdo Grande

Figure Files:

Data Requirements:

Reproducibility:

References

---

## [Editor Report · Decision Letter 1]

13 Nov 2022

Dear PhD Santos,

We are pleased to inform you that your manuscript 'Western blot using Trypanosoma cruzi chimeric recombinant proteins for the serodiagnosis of chronic Chagas disease: a proof-of-concept study' has been provisionally accepted for publication in PLOS Neglected Tropical Diseases.

Best regards,

Andrea Angheben

Academic Editor

Charles Jaffe

Section Editor

Dear authors, many thanks for the work you did to comply with reviewers' comments. Moreover we appreciated your will to update the manuscript concerning the already available blots (LDBio and Abbott). This information and the new evidence you generated are very important to imporve diagnosis of chronic Chagas disease.

---

## [Editor Report · Acceptance letter]

22 Nov 2022

Dear PhD Santos,

We are delighted to inform you that your manuscript, "Western blot using Trypanosoma cruzi chimeric recombinant proteins for the serodiagnosis of chronic Chagas disease: a proof-of-concept study," has been formally accepted for publication in PLOS Neglected Tropical Diseases.

Best regards,

Shaden Kamhawi

co-Editor-in-Chief

Paul Brindley

co-Editor-in-Chief
